# Aqueous Binary Mixtures of Stearic Acid and Its Hydroxylated Counterpart 12-Hydroxystearic Acid: Fine Tuning of the Lamellar/Micelle Threshold Temperature Transition and of the Micelle Shape

**DOI:** 10.3390/molecules28176317

**Published:** 2023-08-29

**Authors:** Maëva Almeida, Daniel Dudzinski, Bastien Rousseau, Catherine Amiel, Sylvain Prévost, Fabrice Cousin, Clémence Le Coeur

**Affiliations:** 1CNRS, ICMPE, UMR 7182, 2 Rue Henri Dunant, Université Paris Est Creteil, 94320 Thiais, France; maeva.ferreira-almeida@u-pec.fr (M.A.); bastien.rousseau@etu.u-pec.fr (B.R.); catherine.amiel-guenoun@cnrs.fr (C.A.); 2Laboratoire Léon Brillouin, Université Paris-Saclay, CEA-CNRS UMR 12 CEA Saclay, 91191 Gif sur Yvette, France; daniel.dudzinski@cea.fr; 3Institut Laue-Langevin, 71 Avenue des Martyrs, CS 20156, CEDEX 9, 38042 Grenoble, France; prevost@ill.fr

**Keywords:** 12-hydroxystearic acid, stearic acid, self-assembly, mixtures, thermo-responsive, micelles

## Abstract

This study examines the structures of soft surfactant-based biomaterials which can be tuned by temperature. More precisely, investigated here is the behavior of stearic acid (SA) and 12-hydroxystearic acid (12-HSA) aqueous mixtures as a function of temperature and the 12-HSA/SA molar ratio (R). Whatever R is, the system exhibits a morphological transition at a given threshold temperature, from multilamellar self-assemblies at low temperature to small micelles at high temperature, as shown by a combination of transmittance measurements, Wide Angle X-ray diffraction (WAXS), small angle neutron scattering (SANS), and differential scanning calorimetry (DSC) experiments. The precise determination of the threshold temperature, which ranges between 20 °C and 50 °C depending on R, allows for the construction of the whole phase diagram of the system as a function of R. At high temperature, the micelles that are formed are oblate for pure SA solutions (R = 0) and prolate for pure 12-HSA solutions (R = 1). In the case of mixtures, there is a progressive continuous transition from oblate to prolate shapes when increasing R, with micelles that are almost purely spherical for R = 0.33.

## 1. Introduction

A current trend in the field of Soft Matter is the design of so-called “smart” stimuli-responsive materials whose macroscopic properties can be tuned over a large range on demand and in a controlled manner. This stems from the fact that the structures of the self-assemblies formed by the elementary bricks of the soft matter systems (surfactants, nanoparticles, polymers, and liquid crystals) can strongly reorganize themselves under the effect of the slight variations in their environment, which in turn tune their macroscopic properties, particularly the rheological ones. The desired macroscopic changes can then be triggered either by a change in composition (pH, ionic strength, etc.), by an external stimulus (light, magnetic field, shear, ultrasounds, etc.), or by temperature or pressure.

In such a context, the possibility to design stimuli-responsive foams or emulsions has attracted a lot of interest lately. The goal is to obtain a fine control of the lifetime of these metastable systems through the physico-chemical parameter(s) chosen to trigger their stability. Depending on the targeted application, this necessitates either arresting or speeding up the three main mechanisms that drive their destabilization, namely the Oswald ripening, the coalescence, and the drainage. This thus passes by both a fine control of the elasticity of the layer at the interface (air/water in foams or water/oil in emulsions) and by the viscosity of the aqueous phase for the drainage.

As bio-based and renewable molecules, fatty acids, that are carboxylic acids with an aliphatic chain and have been used since antiquity as surfactants, are the tools of choice for the conception of such stimuli-responsive foams or emulsions. The first way to play vary the Krafft temperature, as the formation of a stable non-aqueous foam via the Pickering approach, has been achieved by the irreversible adsorption of fatty acids crystals below their melting temperature [1,2], following a concept originally developed on SDS foams [3]. However, the most common use of fatty acids is when they are solubilized in aqueous solutions below their Krafft temperature, which necessitates to lower it in the case of fatty acids with long saturated chains (C14 to C22) that are not soluble at room temperature and has conventional metallic counterions. Such a solubilization step can be achieved by the use of organic counterions, usually an alkyl amine, that makes ion pairing with the carboxylated heads [4,5,6] or with cationic surfactants that form catanionic self-assemblies [7]. In the case of organic counterions, the main physico-chemical parameters that tune both the properties of the monolayers and the morphologies of the self-assembled aggregates in aqueous solutions are the pH and the temperature. Both play on the packing parameter of fatty acid molecules. The pH controls the ionization state of the carboxylated heads, and therefore, the competition between the H-bonding of the protonated COOH heads and the electrostatic repulsions between the deprotonated COO- heads. For its part, temperature controls the state of crystallinity of the fatty chains. Recent studies have shown that the self-assembled aggregates based on the long fatty acid chains display a broad polymorphism as a function of these two parameters, mainly lamellar at a low pH and/or low temperature (facetted 2D objects, planar lamellar phases, and unilamellar or mutilamellar vesicles) [8,9,10,11] and spherical micelles at a large pH and/or temperature [8,9,10,11,12,13,14], with wormlike aggregates in an intermediate regime. Stimuli-responsive foams or emulsions based on the long fatty acid chains have thus mostly relied on these two parameters [15]. In the case of pH, its modification often goes through the addition of an additional species within the liquid phase, which is problematic once the foams or emulsions are formed. This drawback can be circumvented for fatty acids in the presence of CO_2_ (gas) that may decrease the solubility of stearic acid and have recently given rise to the design of many CO_2_-stimulable systems [16,17]. pH changes, and therefore foam stability, can also be triggered via the UV irradiation of a photoacid generator added in aqueous solutions from the beginning [18].

Compared to saturated ones, hydroxylated fatty acids with long chains have a huge potential since the hydroxyl function along the alkyl chain brings three useful properties to the molecules. First, a second hydrophilic center that enable them to adopt an unconventional conformation at the air/water interface if the OH group is located sufficiently far from the head group [19,20,21]. Second, the possibility to make a hydrogen bond network within the monolayer and to therefore make it very elastic [21]. Third, it makes them chiral, which enables the formation of self-assemblies with unconventional morphologies in aqueous solutions, such as twisted ribbons [22] or multi-lamellar tubes that come from the rolling of helical ribbons in the case of 12-hydroxy stearic acid (12-HSA) [23]. Such a structure has also been recently described in the literature from the complexes of ionic surfactants and cyclodextrins [17,24,25,26,27].

These multi-lamellar tubes of 12-HSA are made of a few (~3–6) stacked bilayers separated by a lamella of water. These lamellas roll up in the tubes of a micrometer diameter size and length of 10 µm or more. The tubes swell over a wide range of temperatures. They can be obtained from a large set of counterions [28,29], in various physicochemical conditions [30], and they melt into small micelles with a threshold melting temperature [28] that depends on the counterion/12-HSA ratio [29]. The transition from multi-lamellar tubes to micelles makes them a tool of choice for the design of thermo-responsive systems since the viscosity in the tubes’ regime is larger by 3 to 5 orders of magnitude, depending on swelling, than that of the regime of small micelles at high temperature [29,31]. This has enabled the design of the first ultra-stable and responsive aqueous foams reported in the literature [15], the drainage being started on demand from the drop of the viscosity. The temperature of destabilization is tunable via an appropriate choice of the type of counterion and the counterion/12-HSA ratio [32]. Additionally, the properties of such fatty acid assemblies in homogeneous one-phase systems could also be really interesting. It has been demonstrated that the 12-HSA tubes can be loaded via molecules of biomedical interest, which also makes them very appealing for the design of drug-release-triggered systems [33]. Small hydrophobic drugs could be encapsulated in thermo-responsive self-assembled structures in the same way they could be encapsulated in thermos-stimulable polymeric micelles like pluronic micelles, for instance [34,35].

One step further for the design for the thermostable long-fatty acids chains-based architecture would be the design of a system that would be responsive to both CO_2_ and temperature and for which both the rheological surface properties of the monolayers and the rheological bulk properties of the self-assemblies in aqueous solutions would be tunable. With this in mind, we have recently studied mixtures of 12-HSA and its counterpart stearic acid (SA) that differ only by the hydroxyl on the 12th carbon of the C18 alkyl chains of 12-HSA, so that the heads have a single state of ionization, which is a prerequisite for a further triggering by pH. Both are bio-sourced fatty acids which could present a large number of applications. The monolayers of mixtures have a rich behavior since their 2D phase, established by surface pressure versus surface area isotherms, shows a eutectic point with a good miscibility [36]. Even if the surface rheological properties of the mixed monolayers have not yet been established to the best of our knowledge, it is likely that their surface elasticity will be largely dependent on the ratio between the two fatty acids owing to the hydrogen bonds that can be formed between the 12-HSA molecules. In aqueous solutions, when the mixtures are solubilized via ethanolamine counterions, we have shown recently that the self-assemblies display a very broad polymorphism depending on the ratio between the two fatty acids [37]. When doping the SA molecules via a low content of 12-HSA molecules (less than 25%), the planar lamellar self-assemblies that are formed in pure SA solutions turn into complex multi-lamellar-facetted structures with planar domains bounded by the spherical curved domains that may result from a local phase separation. When doping the 12-HSA molecules via a low content of SA molecules (less than 25%), the multi-lamellar long tubes that are formed in pure 12-HSA systems are preserved, but the length of the tubes decrease upon the addition of SA. At the intermediate ratios, spectacular long 1-D multilamellar objects made of tubes bounded by the helical ribbon solutions are formed. The transition between the different regimes of morphologies is accompanied by a change in their viscoelastic properties. While our previous study [37] highlighted the importance of hydrogen bonding on the structures of the mixtures at room temperature, here we focus on the influence it has on the thermo-stimulation of such mixtures on both the transition thresholds and the structures formed at elevated temperatures. We extend here that this is the first study to examine a wide range of temperatures in order to elaborate the whole phase diagram of the SA/12-HSA system, focusing on the transition between the lamellar self-assemblies and micellar assemblies, coupling macroscopic observations, transmittance measurements, Differential Scanning Calorimetry (DSC), Wide Angle X-Ray scattering (WAXS), and Small Angle Neutron Scattering (SANS).

## 2. Results

### 2.1. Macroscopic Behavior and Determination of Threshold Temperature

As in reference [37], we focus on the aqueous mixtures of 12-HSA and SA molecules with an overall content of 2 wt% of fatty acids that are solubilized in water thanks to the use of ethanolamine as counterion with a molar ratio surfactant/counter ion *r* of 0.2. The ratio of 12-HSA to the total concentration of fatty acids R is defined as R = n_12HSA_/(n_SA_ + n_12HSA_) and is probed over the full range varying from R = 0 to R = 1. As described in the Materials and Methods section, the protocol enabling the solubilization of the stock solutions of both SA and 12-HSA in water in the presence of ethanolamine always involve a step where the fatty acid/ethanolamine mixture is heated at 70 °C. The pH of the samples is 10.85 ± 0.15 in H_2_O, meaning that all the fatty acids bear a carboxylate head [37]. The solutions are then macroscopically limpid with an apparent viscosity close to that of water. This behavior is consistent with the literature that shows that either SA [8] or 12-HAS [28] are self-assembled in small micelles at such temperature. The stock solutions are then mixed at 70 °C to reach the targeted R. The mixtures stay limpid and not viscous after this final mixing step for every R under scrutiny.

When the mixtures are cooled down to 20 °C, they undergo a transition from limpid to turbid, along with an increase in viscosity, at a given temperature that depends on R. The turbidity and change in viscosity arise from the formation of the multilamellar self-assemblies of different geometries (planar lamellar phases, multilamellar tubes, or supramolecular assemblies containing both of them) that are large enough to scatter visible light [37]. The temperature of the transition threshold between the limpid and turbid phases, as identified via visual inspection, depends however strongly on R. In order to determine it and establish the macroscopic state diagram of the system, we have measured the transmittance *T* of the samples at 450 nm on a UV-Vis spectrometer as a function of temperature. Whatever the sample, *T* does not decrease sharply from 100% to 0% at a well-defined temperature upon cooling, but over a temperature range of several degrees with a magnitude that depends on the cooling rate. The lower the cooling rate, the sharper the transition (compare Figure 1A, which is obtained with the ramp of 0.2 °C/min, with Appendix A, which is obtained with the ramp of 1 °C/min). Additionally, the onset temperature of the appearance of turbidity is as high as the cooling rate is low. The measurement of *T* over a cooling/heating cycle also shows a strong hysteresis with an onset of the appearance of turbidity at a lower temperature during the initial cooling step than for the following heating step (see Appendix A). It thus takes an important amount of time for the system to reach a steady state close to equilibrium. The transition thus appears to occur over an extended range of temperatures where some self-assembled objects that scatter light coexist with some other ones that do not. The measurements obtained at the lowest cooling rate allow us to build a state diagram for the system that is divided into three regions (Figure 1B): at low temperature, the turbid phase; at intermediate temperature, the regime where the phase transition takes place; and the translucent phase at the high temperature. It is likely that the intermediate region of coexistence is only a kinetic feature and would vanish if the state diagram was built with an infinitely slow cooling rate. The turbid/transition threshold corresponds to the temperature where the transmittance *T* falls to 0, and the transition/translucent threshold corresponds to the temperature where *T* starts to decrease from 1.

For the pure HSA solution (R = 1), the onset of the appearance of turbidity occurs at 35 °C, in accordance with reference [29] where the temperature of the micelles/multilamellar tube transition was determined at such temperatures for an *r* of 0.2. The temperature threshold occurs by contrary at a much larger temperature close to 51 °C for the pure SA system at R = 0, which is consistent with the value obtained by Xu and collaborators [8] in which the reported temperature of the transition is 50 °C. In between, three different regimes can be identified as function of R. At a large content in SA (from R = 0.05 to R = 0.25), the transition temperature decreases slightly and continuously from the R = 0 case upon the progressive introduction of the HSA molecules, with an overall decrease of about 7 °C. Symmetrically, at a large content in HSA (from R = 0.9 to R = 0.75), the transition temperature slightly decreases upon the progressive introduction of the SA molecules; with an overall decrease of about 10 °C from the R = 1 case. At intermediate R, the transition temperature drops strongly upon the increase in R, with a decrease of ~15 °C from R = 0.25 to R = 0.4–R = 0.6, in order to bridge the two regimes of the temperature threshold where one of the two fatty acids is in large excess. It is worth noting that the phase transition occurs over a much larger temperature range for such regimes of the intermediate ratio R where it is more difficult for the system to reach equilibrium.

### 2.2. DSC Measurements

The temperature of the transition threshold was also assessed via DSC measurements. Measurements were performed with two different ramps of 0.2 °C/min and 1°/min. We represent the results obtained during cooling at 1 °C/min (Figure 2) for which thermal events are easier to evidence than those obtained at 0.2 °C/min that have a poorer resolution due to experimental limitations (Appendix A). As already explained for the transmittance measurements, there is a small temperature offset between both sets of the data associated with the slow kinetics that is taken by the system to reach equilibrium. The temperature of the transitions for the two cooling rates are represented in Figure 1B. For the sake of comparison with the transmittance, we have chosen to represent here the data obtained upon cooling. Heating ramps have also been performed (Appendix A) and evidence a large hysteresis between the cooling and heating cycles, similar to the transmittance measurements.

For R = 1 (pure 12-HSA solution), there is one endothermic peak between 30 °C and 35 °C whose shape and width is similar to those obtained in reference [7] for an ethanolamine/fatty acid ratio *r* = 0.2, where it showed that it is associated with both the Lα–Lβ fluid–gel transition of the fatty acids and the transition of the multi-lamellar tubes into micelles. For such an *r*, when all heads are on their COO^−^ form, the melting of the fatty acid chains induces the morphological transition by changing the packing parameter. For larger *r*, when parts of the heads are on their COOH form, it was shown that the Lα–Lβ fluid–gel transition and the melting of the multi-lamellar tubes into micelles gives rise to the two distinct endothermic peaks on the enthalpograms because the H-bonds that remain within the fluid fatty acids above the Lα–Lβ fluid–gel transition have to be broken to allow the transition from the tubes to micelles.

For R = 0 (pure SA solution), we identify a broad peak between 40 and 45 °C (Figure 2). This is consistent with the previous results by Xu et al. [8] for pure SA solutions solubilized via ethanolamine that evidenced a lamella–micelles transition at around 42 °C via coupled DSC and cryo-TEM measurements at a SA concentration (1.5 wt%) close to that of our current study (2 wt%).

In the case of the mixtures, there is only one broad peak whenever the R is under scrutiny, except for R = 0.75. This suggests for all these ratios R that the presence of a single transition corresponds to the melting of both the fatty acids and mesophases by analogy to the R = 1 and R = 0 cases. When R varies, the intensity of the peaks and their width evolves. The peak broadening is probably associated to the time necessary to reach equilibrium, that differs from one R to another, as we chose to measure all the samples with the same temperature ramp to compare their behaviors in similar conditions. For some ratios, the melting temperature can be shifted up to 5 °C by reducing the ramp at 1 °C/min down to 0.2 °C/min. Indeed, the time necessary to reach a turbid steady state after crossing the limpid–turbid transition is very dependent on the ratio R, as shown by the turbidity measurements. At the intermediate R, when the peak becomes hardly visible in the DSC experiment, it is much slower than for the other R.

Three different regimes can be identified depending on R. Firstly, at low R (R = 0, R = 0.05, R = 0.1, R = 0.15, and R = 0.25), the peak decreases in intensity and widens at the same time. The transition temperature decreases slightly from 42 °C for R = 0 to 36 °C for R = 0.25. At R = 0.25, the peak is hardly detectable at a ramp of 1 °C/min and not detectable at 0.2 °C/min. This may be due to a broadening of the melting peak to such an extent that the apparatus is no longer sensitive enough to enable its measurement. At intermediate R (R = 0.4, R = 0.5, and R = 0.6), there is a large peak between 15 °C and 20 °C whose intensity increases with R. At high R (R = 0.9 and R = 1), where the mixtures form multi-lamellar tubes at 20 °C [37], a rather broad but well-defined melting peak is observed whose intensity increases with R. The transition temperature also increases when the proportion of HSA increases. For all these ratios, the fact that there is only one single transition, as in the case of pure 12-HSA at *r* = 0.2, suggests that it corresponds to both the transition Lα–Lβ and the multilamellar–micellar transition. For R = 0.75, we observe two different peaks. The comparison with the other sample at different R tends to indicate that there are two different structures that are melting at different temperatures in the sample. At ambient temperature, we already noticed that the solution is a mixture of two different fatty acid architectures for such ratios [37], with multi-lamellar tubes co-existing with multi-lamellar ribbons. We hypothesize that the sharper peak at 26 °C corresponds to the melting of the multilamellar tubes, similarly as for the higher R, whereas the broad peak at 20 °C may correspond to the melting of the ribbons.

The temperatures of transition obtained via DSC match the threshold temperatures obtained via the transmittance measurements, as shown in Figure 1B. It demonstrates that the thermodynamic transition evidenced via DSC corresponds to the melting of the large, self-assembled structures.

### 2.3. Wide Angle X-ray Scattering (WAXS)

WAXS measurements were performed to determine if the fatty acid chains within the bilayers are fluid or in a gelled state for the whole range of the HSA/SA mixtures at 36 °C and 50 °C (Figure 3). For the sake of comparison, they are compared with the data at 20 °C that we already obtained in [37] that showed some Bragg diffraction peaks and demonstrates that the fatty acids are in a crystalline gel state in all cases. At 50 °C, whatever R is, the diffractograms no longer show these Bragg peaks, revealing a completely fluid assembly of the different fatty acids. At 36 °C, the diffractograms show either the signature of the crystalline fatty acid self-assembly with the presence of a Bragg peak, or those of the fatty acids in a fluid state as observed at 50 °C, depending on the ratio R.

Three different regimes of R can be distinguished. At low R, from R = 0 (pure SA) to R = 0.25, a Bragg peak is observed at 20 °C for q_peak_ = 1.534 Å^−1^. As the temperature is increased to 36 °C, it is still present, but this position is shifted towards lower q (q_peak_ = 1.520 Å^−1^). When there is a majority of SA in the mixture, the main crystalline structure is one of the pure SA samples at both 20 °C and 36 °C. However, there are noticeable changes in the crystalline structures formed when the temperature is increased. This is likely due to a change in the SA crystalline structure, not disturbed by the addition of a small amount of HSA molecules. At 50 °C, all samples no longer present any crystalline peaks. The crystal–fluid transition of the fatty acids thus occurs between 36 °C and 50 °C in this regime, exactly in the temperature range where the endothermic peak was observed with the DSC measurements. This indicates that the transitions evidenced in the enthalpograms correspond to the crystal–fluid transitions. As R increases (for R from 0.4 to 0.6), the crystalline structure obtained at 20 °C with the Bragg peaks of low intensity, i.e., crystalline domains of low size, disappear at 36 °C and 50 °C. This is again consistent with the temperature transitions obtained via DSC. There are smaller crystalline parts in the mixed fatty acid mixtures. For R ≥ 0.75, the two Bragg peaks present at 20 °C, similar to those of the crystalline phase of the fatty acids in the pure 12-HSA sample (R = 1) forming the multilamellar tubes, are no longer observed at 36 °C and 50 °C, in accordance with the temperature transition thresholds (Figure 1B) that are all below 35 °C for this regime of R.

In summary, as previously observed for the pure SA and 12-HSA solutions, the crystalline structures of the gelled bilayers at 20 °C of the mixed solutions melt between 20 °C and 50 °C depending on R.

### 2.4. Structure of Samples by SANS

The structure of the self-assemblies was determined via SANS. All the samples were prepared in D_2_O while the data obtained with the other experimental techniques were obtained with the samples prepared in H_2_O. This isotopic exchange would possibly shift the transition temperatures. We have checked that this is not the case via transmittance measurements (Appendix A) that demonstrate that the transition temperatures are similar in the D_2_O and H_2_O solutions.

All the results are represented in Figure 4 for each temperature (30, 37, 45, and 60 °C). Depending on R and/or the temperature, the scattered intensity shows three different behaviors: (*i*) at low temperature, when the samples are turbid white, the scattering is characteristic of multilamellar self-assemblies; (*ii*) at high temperature, for translucent samples, the scattering is characteristic of small micelles interacting via repulsive interactions, and (*iii*) in the transition domain at intermediate temperature, the scattering is typical from the combination of the scattering of the two types of objects. The turbidity arises thus from the formation of multi-lamellar objects that have a sufficiently large size to scatter light. When all or most of the surfactant molecules are involved in small micelles, the sample is translucent and starts becoming partially turbid as soon as a few molecules make the lamellar objects. The translucent/turbid transition threshold of the state diagram thus corresponds to the onset of formation of these first multi-lamellar objects.

At low temperature, we recover the scattering of the different multilamellar self-assemblies that were obtained at 20 °C and exhaustively described in reference [37]. For all samples, there are several Bragg peaks associated with the interlamellar distance at intermediate q with an oscillation at q~0.25 Å^−1^ originating from the lamella form factor at large q. Only the low q part differs from one self-assembled structure to another: q^−2^ for planar lamella at R = 0, q^−4^ at low R for the lamellar-facetted structures with planar domains bounded by the spherical curved domains, q^−3^ for the mixtures of helical ribbons and multilamellar tubes; and q^−3^ with an oscillation-associated radius of the tubular shape for the multilamellar tubes. The scattering curves were thus fitted in the intermediate and large q region following the same approach we used in reference [37] via a model proposed by Nallet et al. [38] to determine the structural parameters of lamella (thickness, d-spacing, and rigidity, which is represented in Figure 5). It considers a form factor of the lamella and a structure factor between lamellas for which both the number of stacked bilayers and the Caillé parameter (η), accounting for the thermal fluctuations of the bilayers, were adjusted. This model enables to satisfactorily fit all the data for which multi-lamellar objects were formed, respectively, for all R except from 0.4 to 1 at 30 °C (Figure 4A), all R ≤ 0.25 at 37 °C (Figure 4B), and all R ≤ 0.15 at 45 °C (Figure 4C). Although the ratios 1, 0.9, and 0.4 at 30 °C present some Bragg peaks that refer to multi-lamellar objects, at intermediate q, they start concomitantly to present the curved shape of a micelle, thus showing that the solutions contain both lamellar objects and micelles. For this reason, these scattering curves were not fitted by the sum of a Nallet model for the lamella and interacting micelles, as there would be too many free parameters in the modelling to provide an unambiguous result.

All the results of these fits are represented in Figure 4. The low q part of the curves was not fitted as it displays different behavior from one sample to another (q^−2^ versus q^−3^ versus q^−4^). Please note that the results obtained at R = 0.4 were not analyzed. Indeed, we showed in [37] that the sample at 20 °C was not homogeneous at the millimetric scale at 20 °C with a possible phase separation, which gives a strong uncertainty on the effective concentration that is actually probed by the neutron beam.

Whatever R is, the increase in temperature does not change the morphology of the self-assembly for a given R. As was observed in the case of pure 12-HSA solutions in [28], the increase in temperature is accompanied by a slight decrease in the inter-lamellar distance (Figure 4A), while the Caillé parameter does not evolve much with temperature except for the ratio R = 0.25 (Figure 4B). The thickness of the lamella does not evolve with the temperature and remains constant around 23 Å for each of the R ratios. This thickness is close to the size of a fatty acid (21 Å) [29] and corresponds to the interdigitated lamellar phases. This result is noticeable for the samples with low R (R ≤ 0.25). Indeed, when the self-assemblies are predominantly formed of SA, WAXS measurements reveal that the lattice parameter of the crystallized fatty acids evolve with temperature. This local modification of the structure is not accompanied by a significant change in the thickness of the lamella. Structural reorganization thus occurs in-plane.

Let us now discuss the scattered curves obtained for all R ≥ 0.25 at 45 °C and for R = 0, R = 0.1 and R = 0.15 at 60 °C, for which the Bragg peaks associated to the lamella self-assemblies have completely vanished (Figure 4C and Figure 4D, respectively). They are completely different from those at 20 °C and are typical of the one of small micelles interacting via repulsive interactions, which is consistent with the literature. They display three main features: (*i*) at very low q values (q < 0.01 Å^−1^), the scattering intensity decreases when q decreases, showing that the isothermal osmotic compressibility (χ_T_ ) of the system is very weak due to the electrostatic repulsions; (*ii*) at intermediate q, it shows a strong correlation peak at q*, ranging between 0.02 Å^−1^ and 0.06 Å^−1^, depending on the sample, that correspond in direct space to the mean distance between micelles (2π/q*); and (*iii*) for q > 0.1 Å^−1^, a Porod-like decay in q^−4^ stemming from the 3D character of the micelles with a marked oscillation at ~0.25 Å^−1^ is associated with the first minima of the form factor.

We assume for the first time that micelles are spheres. Since they are centrosymmetric objects, the scattering intensity *I*(*q*) can be written as shown below:(1)$$I\left(q\right)=\Phi {\left(\rho_{fatty\_acid}-\rho_{D2O}\right)}^{2}VP\left(q\right)S\left(q\right)$$
where *Φ* is the volume fraction of micelles, ρ_fatty_acid_ and ρ_D2O_ are the respective neutron scattering length density of the fatty acids (considering in the first approximation that it is similar for 12-HSA and SA) and D_2_O, V is the volume of the micelle; *P*(*q*) is the normalized micelle form factor; and *S*(*q*) is the inter-micelles structure factor.

The low compressibility of the samples, associated with a strong correlation peak that reveals a homogeneous distribution of the self-assembled objects within the sample, highlights the presence of the strong repulsions between micelles. Since the pH of all the solutions is around 10.85 ± 015, each fatty acid bears a carboxylate head. The micelles are thus strongly negatively charged, and the repulsions are of electrostatic origin. The structure factor *S*(*q*) can thus be fitted via a model proposed by Hayter and Penfold for charged 3D objects [39] that considers a Yukawa potential describing electrostatic interactions following the DLVO theory in the mean spherical approximation [40], as will be shown later on.

At large q, *S*(*q*)*_q→∞_* = 1 by principle. In practice, *S*(*q*)~1 is here for q > 0.1 Å^−1^, which enables the decoupling of the structure factor and the form factor during the fitting procedure in order to obtain an accurate description of the form factor without any assumption of the structure factor.

Let us now describe the influence of R on the micelle shape. For R = 1 (pure 12-HSA solution), most studies in the literature have focused on suspensions with a stoichiometric ethanolamine/fatty acid ratio (*r* = 0.5) [29]. For this R, the HSA micelles form spherical micelles with a radius of 22 Å at high temperature, which corresponds to the length of the hydrophobic tail [29]. We have tested such a form factor model of spheres in our case (*r* = 0.2), and the best fit gives a micelle radius of 18.5 Å. This result agrees with the literature [29] at such *r* of 0.2 at 72 °C, but the curve obtained does not properly fit the data.

We therefore chose to refine the model and fit the scattered intensity of the 12-HSA solution at 45 °C using a homogeneous ellipsoidal micellar model that is both compatible with the observation of a characteristic length of 18.5 Å and the size of the molecule. For a dispersion of the ellipsoids of revolution, the decoupling approximation of the structure factor and of the form factor is not valid. It has however been shown by Green et al. [41] that the approximation stays valid for either a dilute dispersion and/or the low aspect ratios of the ellipsoids. This is the case for the ranges of the volume fraction under scrutiny in our study (~2%) and the aspect ratios (between 0.5 and 1.7, see later), with the deviation of the decoupled *S*(*q*) being less than 1% from an explicit calculation of *S*(*q*) of the hard ellipsoids (see in Figure 2 of reference [41]). We will then postulate that the decoupling approximation is valid for all data on micelles presented afterwards. The model refines our data well by giving an equatorial radius of 18.5 Å and a polar radius of 32 Å, i.e., an ellipticity of 1.7. HSA micelles are therefore prolate micelles. The fatty acids are thus not interdigitated within micelles, contrary to the lamellar phase.

For R = 0 (pure SA solution) at 60 °C (Figure 4), a simple look at the evolution of the intensity with respect to the 12-HSA case shows that the SA micelles have a larger aggregation number *N* than the 12-HSA ones (see Appendix A that compare all curves in cm^−1^ without any multiplication by a pre-factor). Indeed, the q-position of the structure correlation peak *q** shifts towards low q for R = 0. Since *q** is inversely proportional to the distance between scattering objects, this shows that the distance between micelles is larger for R = 0 than for R = 1. Given that the total fatty acid concentration in the solution is constant for both ratios, this implies that *N* is larger for R = 0. Moreover, at the same time, the q-range where the Porod q^−4^ arises shifts towards low q for R = 0 with respect to R = 1. As for the pure 12-HSA micelles, it was not possible to properly fit the scattering curve by using a purely spherical form factor, and the best modelling was obtained with an elliptical form factor, with a polar radius of 18.9 Å and an equatorial one of 38.5 Å. The shape of the SA micelles is thus oblate, which is strikingly different from the 12-HSA case where they are prolate.

For the other R, the evolution of the intensity as a function of R shows that the aggregation number of fatty acid per micelle *N* decreases varies continuously from one sample to another (see Appendix A), from R = 0 to R = 1. Indeed, q* shifts towards large q with an increase in R, and the q-range where the Porod q^−4^ arises shifts towards large q at the same time. The data were fitted at large q via an ellipsoidal micelles model [42] (detailed in Appendix A) for all R to account for the evolution of the shape. Polar and equatorial radiuses are represented in Figure 6A and the ellipticity in Figure 6B. The ellipticity *e*, defined as 1-(radius polar/radius equatorial), decays linearly when R decreases and takes values lower than 1 for R < 0.25. The system transitions from prolate micelles at high R, similar to the pure 12-HSA micelles, to oblate micelles at low R, similar to the pure SA micelles. Spherical micelles are obtained when R is close to 0.33. The ellipticity can be tuned from the choice of R with *e* = 0.58 + 1.2 R.

The aggregation number, *N*, was determined in three different ways: one deduced from the mass of the scattering objects, from the extrapolation of the scattered intensity when q tends to 0 I(q) _q→0_, where the structure factor equal to 1 on the full q-range, which we call *N*_Weight_; one obtained by the ratio between the volume of the ellipsoid and that of a fatty acid molecule, which we call *N*_volume_; and one extracted from the position of the correlation peak, *q**, assuming that the repartition of micelles is spatially homogeneous, i.e., the mean distance between micelles is 2π/*q**, that we call *N_q*_*.
(2)$$N_{weight}=\frac{N_{a}I_{0}}{V_{fatty\;acid}^{2}.\;∆\rho^{2}.c.M_{fatty\;acid}}$$
where $V_{fatty\;acid}$ is the specific volume of the fatty acid, Δ*ρ* is the neutron-scattering contrast between the fatty acid and D_2_0 and *c* the concentration of both fatty acids, *N*_a_ is the Avogadro number, and *M_fatty acid_* is the average molar mass of the fatty acids at each ratio (i.e., *M_fatty acid_* = R.M_HSA_ + (1 − R) × M_SA_).
(3)$$N_{volume}=\frac{V_{ellipsoid}}{V_{fatty\;acid}}$$

The volume of a molecule of the fatty acid is approximated by using Tanford’s formula for the alkyl chain volume (V = 27.4 + 26.9n_c_, where n_c_ is the number of the alkyl chain carbons) [43]. We have chosen to use the same value whatever the ratio R is, with the slight error introduced in this calculation being minute, as shown below.
(4)$$N_{q*}=\frac{\Phi_{v}.{\left(\frac{2\pi }{q^{*}}\right)}^{3}}{V_{fatty\;acid}}$$
where $\Phi_{v}$ is the volume fraction in the fatty acid and *q** is the position of the correlation peak. The *N*_weight_ and *N*_volume_, *N_q*_* values have the same order of magnitude and vary in the same way. At high R, the values obtained from the three approaches are slightly different, which may be due to the value of the volume of a 12-HSA molecule, which might be slightly different from the one of the pure SA used in this calculation for the uncertainty of the scattering length density of the micelles as the exact localization of the counter-ion is unknown and also for the uncertainty on the different fits. At R = 1, *N* is close to 100 and varies only slightly when decreasing R up to R= 0.5. It then strongly increases and reaches values between 250 and 350 molecules at R = 0, depending on the method used to obtain it. The calculation of *N_q*_* is not fit-dependent and is probably more accurate than the others. In particular, it does not depend on the scattering length density of the core of the micelle that we have considered equal to that of a fatty acid in the fitting, a value that may be slightly wrong due to the presence of the ethanolamine counterions.

The experimental structure factors were then obtained by dividing the scattered intensity via these adjusted form factors (Equation (1)). They are represented in Appendix A, which allow us to highlight the very low compressibilities of the suspension, and therefore, the strong repulsions. They were fitted with a model by Hayter and Penfold [39], fixing all the parameters to their known experimental values (temperature, ionic strength calculated from all ions introduced in the solution, concentration, and dielectric constant), except for the effective charge by micelle *Z*_micelle_. *Z*_micelle_ varies from 20 to 50 per micelles depending on R and micelles’ size (see Appendix A). The large values obtained for the effective micellar charge are in agreement with the charged character of the fatty acids.

## 3. Discussion

The corpus of results gathered from the different experimental techniques (transmittance, WAXS, SANS, and DSC) converges to a consistent picture and demonstrates that the self-assemblies have the same overall behavior for all ratios R with a transition from a multilamellar phase to a micellar phase in the solution. Indeed, the temperature of the thermodynamic transition evidenced via DSC matches the onset of the appearance of turbidity, the fluid/gel transition of the bilayer, and the morphological transition at the nanoscale. This allows us to build the phase diagram of the HSA/SA mixtures as a function of the R ratio and temperature (Figure 7).

The fact that the lamellar/micelle transition is driven by the melting of the gelled lamellae of the fatty acids in all mixtures highlights the role of the concentration ratio *r* between the fatty acid and the ethanolamine counter ion on the mechanism of the transition. Indeed, while the solutions of the pure SA show a single lamellar phase/micelle transition regardless of the ethanolamine concentration [8], the 12-HSA solutions undergo either only one transition at a high ethanolamine concentration (*r* = 0.2), in accordance with the findings of the current study, or two distinct enthalpic phenomena at a lower ethanolamine concentration, i.e., a larger *r* (*r* = 0.5) that corresponds respectively to the melting of the lamellae and to the lamellar self-assembly/micelle transition at a larger temperature. The reason invoked to explain such a peculiar behavior at large *r*, when the fatty acids heads are not all charged, was the necessity to break the H-bonds that remains between heads on their COOH form [29]. This is, however, not sufficient to capture all the subtle mechanisms of the transition, otherwise the same behavior would have also been observed for the pure solutions of SA. It has to be associated with the ability of the ethanolamine to penetrate/interact within the bilayer, which must be easier in SA bilayers than in the 12-HSA ones, given that the latter forms H-bonds between fatty chains. Indeed, at *r* = 0.5, the 12-HSA crystallized bilayers are not interdigitated [28], while they are interdigitated at *r* = 0.2 [37]. We demonstrate here that the crystallized bilayers are interdigitated in all mixtures at low *r*, without temperature shift. Thus, such a temperature shift of the lamellar/micelle transition with respect to the fluid/solid transition of the lamella only occurs when the latter are not interdigitated. We also demonstrate here that the fatty acids are no longer interdigitated once melted when they do form micelles.

If all mixtures display the same overall behavior, the ratio R has an influence on both the threshold temperature of the transition and the ability of the system to reach its steady state with time when changing temperatures. At intermediate R, the transition temperature passes by a minimum at R = 0.6. At the same time, the transition temperature zone, where micelles and lamellar phase truly co-exist (see SANS data at 37 °C in Figure 4A), is broad in this regime of intermediate R, with a DSC peak that is very broad. Such a regime of R corresponds to the ratios for which we demonstrated in [37] that there is a partitioning at the local scales between the 12-HSA and SA molecules at 20 °C, which is at the origin of the minimum of the temperature threshold. The sizes of the crystalline domains in this region of R are indeed smaller, as shown by the WAXS data. Given that the melting temperature is a function of the size of the crystal domains, it is logical to have a lowered melting temperature in the mixtures with respect to the pure 12-HSA and SA solutions. It is worth mentioning that the shape of the phase diagram is comparable to the phase diagrams bearing a eutectic point, that would be located at ~R = 0.6. Nicely, this ratio is close to the one where the presence of a eutectic point has been demonstrated by Matuo et al. [36] (R = 0.75) that built the 2D phase diagram of the mixed monolayers of HSA/SA at the air/water interface. In the case of mixed monolayers, this ratio depends on the length of the alkyl chain and is not observed for longer alkyl chains. It is thus interesting to point out that the behavior of the HSA/SA mixtures in monolayers mirrors that of the mixtures in aqueous solutions. It suggests that partitioning probably also occurs in monolayers. The partitioning also explains why the transition zone is larger at intermediate R, since the two types of domains that co-exist in lamellar self-assemblies have to form upon cooling (respectively melt upon heating) within the same sample.

The temperature also influences the morphology of the lamellar self-assemblies formed below the lamellar/micelles transition temperature. For pure 12-HSA solutions (R = 0) at *r* = 0.2, the inter-lamellar distance decreases with the increasing temperature, in accordance with the literature [28]. Such a behavior does not happen for pure SA solutions since the interlamellar distance does not vary with the temperature (see Figure 5A). However, the Caillé parameter as well as the lattice parameter of the crystalline phase slightly varies. The interactions between lamella are thus almost temperature independent, given that they are mainly driven by the electrostatic interactions originating from their negatively charged heads, while the rigidity is directly influenced by the packing of molecules within the layer. In the case of mixtures, the interlamellar distance decreases slightly with the temperature in all cases, as for the pure 12-HSA case at R = 1, even for the mixtures containing low amounts of 12-HSA. The presence of 12-HSA molecules thus plays a prominent role on this inter-lamellar distance, whether partitioning occurs or not, and whatever the crystalline structure of the fatty acids is within the lamella.

Let us now discuss the influence of the ratio R on the structure of the micelles that are formed above the lamellar/micelles threshold temperature. This the first time to the best of our knowledge that it is demonstrated that 12-HSA micelles at *r* = 0.2 are ellipsoidal prolate and not spherical, as many surfactant micelles are [44,45,46]. The most well-known case is that of cetyltrimethylammonium bromide (CTAB) micelles [47] with an aspect ratio that varies largely as a function of either the ionic strength [47] or when mixed with other small molecules [48,49,50]. In different surfactants mixtures such as the cocamideopropyl betaine (CAPB)/sodium dodecyl sulfate (SDS) binary system or the dodecanoic/sodium Lauryl ether sulfate/cocamideopropyl betaine (CAPB) ternary system, a more drastic transition from spherical to rodlike micelles or from wormlike to disklike micelles has also been observed [51,52]. It is therefore not surprising to observe that HSA micelles, spherical for *r* = 0.5 are prolate when ethanolamine is added at *r* = 0.2. Along the same lines, we also demonstrate for the first time that SA micelles are oblate. The aspect ratio of a micelle is keen to change when a surfactant is mixed with another surfactant, as is the present case for all mixtures. The ellipticity varies continuously when the ratio R is changed. In an excess of 12-HSA, in the regime of high R, the progressive addition of SA decreases the ellipticity of the micelle while maintaining its prolate shape. One can hypothesize, as schematically represented in Figure 6E, that SA molecules are distributed randomly within the 12-HSA molecules and are slightly modified in the average curvature of micelles. Symmetrically, when some 12-HSA molecules are introduced in oblate micelles of SA, the ellipticity increases when increasing R from R = 0 towards more spherical micelles. In the regime of low R (R < 0.25,) we have shown in [37] that there is a partitioning between the SA and 12-HSA molecules driven by the hydrogen bonds within the facetted lamellar self-assemblies formed at 20 °C originating from the formation of the H-bonds between the OH functions on the alkyl chains of the 12-HSA fatty acids. Such partitioning could also occur within the micelles at high temperature, but it is unlikely since the formation of H-bonds is not favored at large temperatures. We have, however, no direct proof of such possible partitioning within the mixed micelles and have chosen to represent a random distribution of the fatty acids within the micelles at high temperature for the sake of honesty (Figure 6E and Figure 7).

## 4. Experimental

### 4.1. Materials and Sample Preparation

The 12-Hydroxystearic acid (HSA) was acquired from Xilong Chemical Co., Ltd., Shantou, Guangdong Province, China, while Stearic acid (SA) was obtained from Sigma-Aldrich. Ethanolamine was sourced from Aldrich Chemistry with a purity of ≥99.5%. H_2_O was supplied by a Millipore system, and D_2_O was procured from Eurisotop, Saint-Aubin, France.

For the preparation of the mother solutions of HSA and SA, each with a concentration of 100 g/L, the respective compounds were weighed and dissolved in ultrapure water to achieve the desired molar concentration of the fatty acids.

The molar ratio surfactant/counter ion *r* (calculated by *r* = n_fatty_acid_/(n_fatty_acid_ + n_ethanolamine_)) was fixed at 0.2. All solutions underwent stirring and were heated to 70 °C in an oven for 2 h, followed by vortexing for homogenization.

Subsequently, mixture samples were prepared with a consistent concentration of fatty acid molecules, set at 20 g/L, by combining stock solutions with ultrapure water in appropriate dilutions. The goal was to achieve the intended ratio of the two fatty acids, denoted as R and defined as follows:$$R=\frac{n_{12HSA}}{\left(n_{12HSA}+n_{SA}\right)}$$
where *n*_12HSA_ and *n_SA_* represent the moles of the respective fatty acids. The mixtures were heated at 70 °C a second time for homogenization. For SANS experiments, all samples were prepared in D_2_O at the same volume fraction.

The paper investigated the following R values: 1 (pure sample of the HSA fatty acids as reference), 0.9, 0.75, 0.6, 0.5, 0.4, 0.25, 0.1, 0.05, and 0 (reference sample of the SA fatty acids). Prior to any measurements, all samples were heated to 70 °C for homogenization.

### 4.2. UV-Visible Spectroscopy (UV-Vis)

UV-Vis experiments were performed on a Varian Cary 100 spectrometer from Agilent (Les Ulis, France). For all the measurements, the wavelength used was 450 nm. We used a thermal program with ramp temperatures between 65 °C and 5 °C (heating and cooling cycles) at a rate of 0.2 °C/min and 1 °C/min.

### 4.3. Differential Scanning Calorimetry (DSC)

The phase transition temperatures were measured on a TA DSC25 calorimeter (Guyancourt, France). For these measurements, we used two stainless steel cells, one containing water used as a reference, and the other one containing approximately 7 mg of the sample. The heating and cooling ramps were between 5 °C and 65 °C at a rate of 1 °C/min and 0.2 °C/min.

### 4.4. Wide Angle X-ray Scattering (WAXS)

WAXS measurements were carried out on a Xeuss 2.0 instrument from Xenocs (Grenoble, France), which uses a microfocused Cu Kα source with a wavelength of 1.54 Å and a PILATUS3 detector (Dectris, Baden, Switzerland). The experiments were performed at a sample-to-detector distance of 350 mm with a collimated beam size of 0.8 × 0.8 mm to achieve a q-range of 0.038–1.78 Å^−1^. The solutions were poured inside 1.5 mm glass capillaries that were placed onto a homemade sample holder thermalized with a circulating water flow coupled with a Huber bath, allowing to control the samples temperature to 36 °C and 50 °C. The measurements were performed for 35 min per sample to achieve a good statistic. The scattering signals from the empty beam, empty capillary, and dark field were measured separately. These signals were then subtracted from the sample scattering, considering their relative transmission. The resulting values were normalized using the incident beam intensity to derive the scattering measurements in absolute units (cm^−1^). Additionally, a reference measurement of the solvent sample was independently obtained and its contribution to the sample scattering was accurately removed.

### 4.5. Small Angle Neutron Scattering (SANS)

Small angle neutron scattering experiments were conducted at the Institut Laue-Langevin—The European Neutron Source (Grenoble, France) using the D11 diffractometer [53]. We employed four different setups (6 Å at 1.7 m, 6 Å 5.5 m, 6 Å at 20.5 m, and 13 Å at 38 m) to cover an extensive q-range from 5.9 10^−4^ Å^−1^ to 0.53 Å^−1^. To enhance the neutron contrast with hydrogenated molecules and minimize incoherent scattering, we prepared the samples in D_2_O. It was verified in advance that substituting the deuterated water for hydrogenated water did not alter the overall appearance of the samples. The samples were contained within flat quartz cells (Hellma, Jena, Germany) with a 2 mm optical path.

All samples were measured at 30 °C, 37 °C, and 45 °C and the samples at R = 0.15 and R = 0.1 were additionally measured at 60 °C. The temperature was set via a circulating bath that thermalized the sample holder into which samples were placed. Transmissions, a scattering of the empty cell, sintered ^10^B_4_C (neutron absorber to estimate the ambient background for the detector), a scattering of the hydrogenated water (as a flat field), and the differential scattering cross section of water (for absolute scale) were measured independently. The subtraction of parasitic contributions and normalization by water to take into account the detectors’ heterogeneities were applied to the raw data via GRASP software (version Grasp Lockdown V.9.22e) to obtain the corrected data in absolute units (cm^−1^) [40]. Solvent and incoherent scattering contributions were subtracted to all spectra.

The fitting software used was SasView 5.0.4 [54]. Fitting models are detailed in the Appendix A.

## 5. Conclusions

The variations in the temperature thresholds that we evidenced for different HSA/SA mixtures suggest that subtle changes in the internal structure of the mixtures can induce significant changes on phase segregations and domain sizes leading to different thermal properties. This observation is important because it highlights the complexity of the interactions in these systems and demonstrates that small changes in the blend composition may enable us to tune the material properties to a large extent. In particular, it opens up interesting perspectives for the design of new functional materials with temperature-tunable properties. Indeed, fatty acid mixtures have been widely studied for their ability to form self-assembled structures such as lamellae and micelles for their further use in a variety of applications including the production of foams and gels [15,32]. The temperature-dependent transition from large self-assemblies to small micelles strongly modulate the rheological properties of these systems, which, depending on the R ratio, go from viscous fluids or gels to solution with viscosity approaching one of water. The establishment of the whole phase diagram of fatty acid mixtures as a function of mixture composition and temperature thus sets the foundations for the design of new fatty acid-based functional materials with stimuli-responsive properties, such as stimulable foams and gels [15,32]. Moreover, the change in self-assembly (from lamellae or multi-lamellar tubes to micelles) may prove particularly interesting for the design of drug delivery systems, where thermo-stimulable properties are particularly sought-after [33].

## Figures and Tables

**Figure 1 molecules-28-06317-f001:**
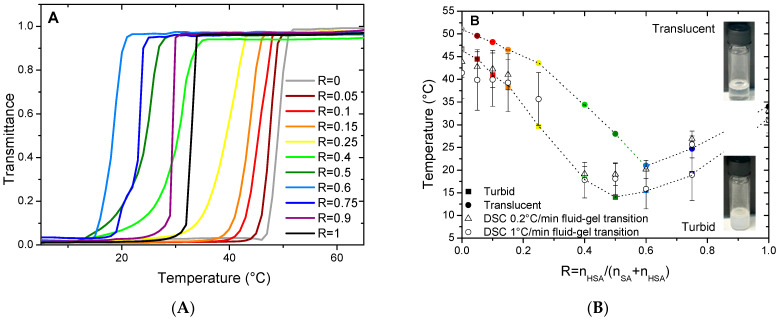
(**A**) Transmittance as a function of temperature for HSA/SA mixtures from R = 0 to R = 1 upon cooling at a cooling rate of 0.2 °C/min. (**B**) Temperatures of transition determined by turbidimetry and by DSC as a function of ratio R, where error bars are estimated from the width of the peak.

**Figure 2 molecules-28-06317-f002:**
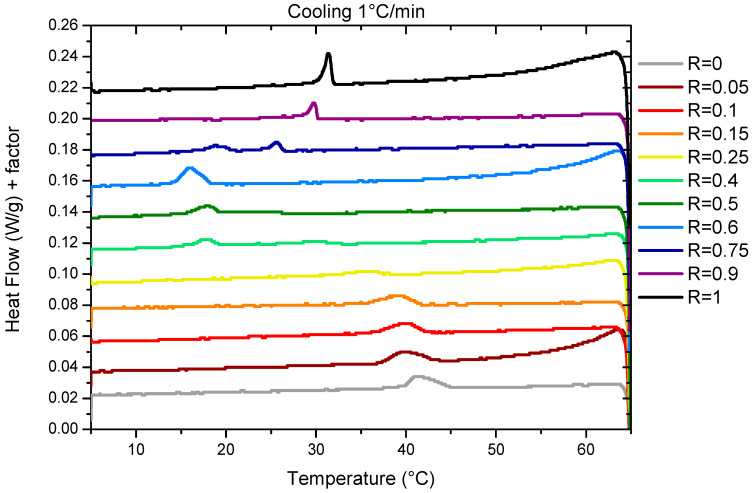
Enthalpograms obtained for 2 wt% mixture in fatty acid at various R ratios upon cooling at a cooling rate of 1 °C/min. Data are shifted for clarity.

**Figure 3 molecules-28-06317-f003:**
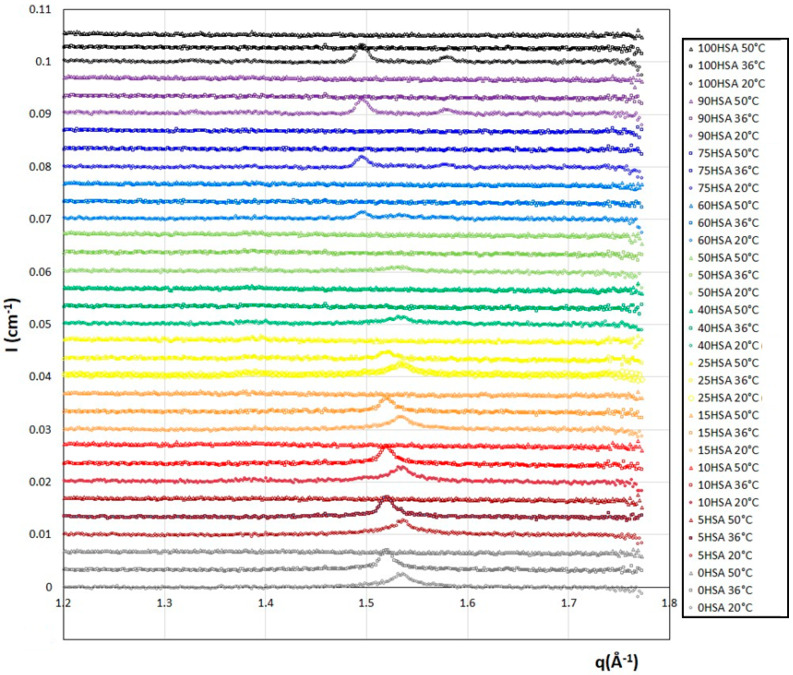
WAXS diffractograms for the different HSA/SA samples ratio, from pure SA (R = 0) to pure HSA (R = 1), at three different temperatures (20 °C from reference [37], 36 °C, and 50 °C). The spectra were successively shifted in intensity by an offset of 0.0033 cm^−1^ for clarity. Data at 20 °C with permission from reference [37].

**Figure 4 molecules-28-06317-f004:**
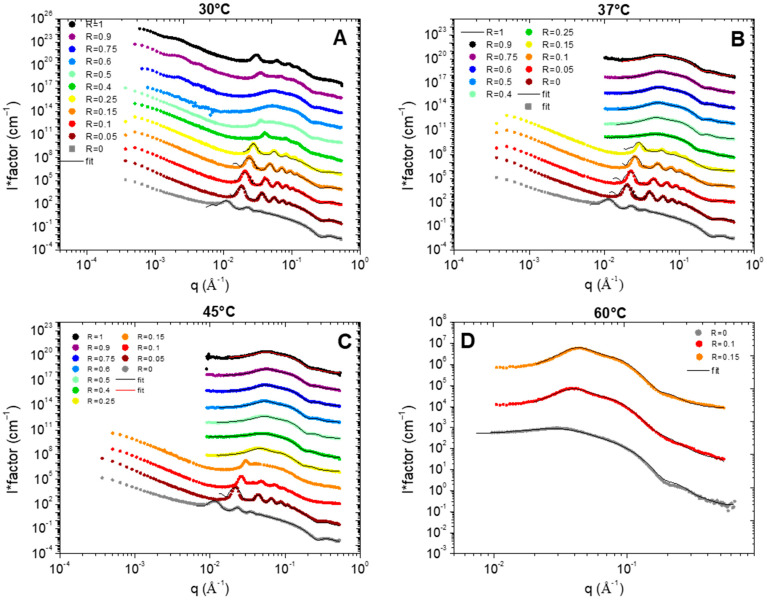
SANS intensity profiles at four temperatures ((**A**): 30 °C, (**B**): 37 °C, (**C**): 45 °C, and (**D**): 60 °C) for the different samples in D_2_O for different HSA/SA ratios, from pure SA (R = 0) to pure HSA (R = 1). The spectra are successively shifted by a factor of 10 in intensity for clarity (data for R = 0 in absolute scale). The black and red continuous lines correspond to the best fit of the data either by a lamellar model or by a model of an elliptical micelles in interactions (see description in Appendix A). Some data showing a combination of scattering of lamellar objects and micelles are not fitted (see explanation in text).

**Figure 5 molecules-28-06317-f005:**
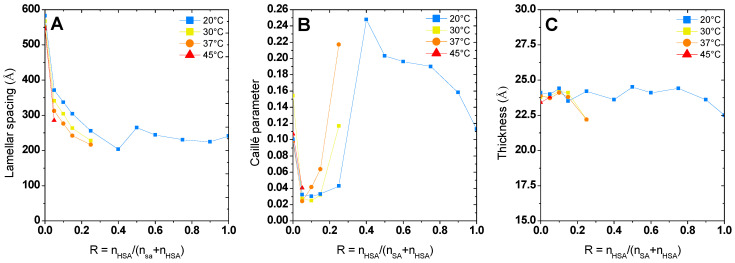
Value of lamellar spacing. (**A**) Caillé Parameter (**B**) and thickness (**C**) obtained from fitting of lamellar phase at different temperature. Data at 20 °C are already published in [37] and adapted with permission from Ref. [37] but are represented for the sake of comparison.

**Figure 6 molecules-28-06317-f006:**
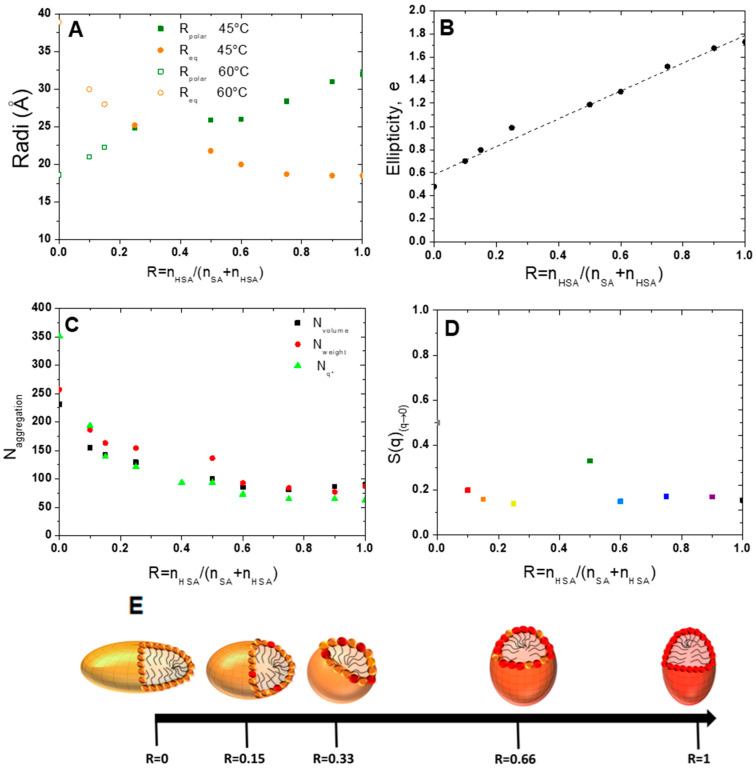
Evolution (**A**) of polar and equatorial radiuses obtained from micelles’ fit at 45 and 60 °C, (**B**) of the ellipticity (**C**) of the number of aggregations *N_volume_* and *N_weight_* as a function of the ratio R. (**D**) *S*(*q*)_(q→0)_ obtained from division on the scattered intensity by the fitted form factor. (**E**) Schematic representation of the evolution of micelles’ shape as a function of the ratio R from prolate micelles to oblate ones. The SA molecules are represented in orange and the 12-HSA molecules in red.

**Figure 7 molecules-28-06317-f007:**
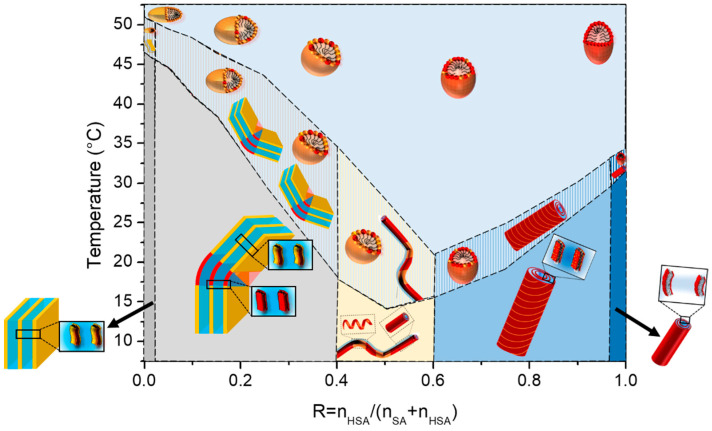
Schemes of the structures of the self-assembled aggregates as a function of R from pure SA (R = 0) to pure 12-HSA (R = 1). The SA molecules are represented in orange and the 12-HSA molecules in red. The total amount of fatty acid is constant and fixed at 2 wt%.

## Data Availability

The raw data will be available from the corresponding authors upon reasonable request.

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
