# Peer review of "Aqueous Binary Mixtures of Stearic Acid and Its Hydroxylated Counterpart 12-Hydroxystearic Acid: Fine Tuning of the Lamellar/Micelle Threshold Temperature Transition and of the Micelle Shape"

_molecules, 2023, doi:10.3390/molecules28176317_

Round 1

Reviewer 1 Report

This is an extremely detailed study of the interesting mixture of two fatty acids that, in aqueous solutions, forms a temperature dependent soft material at room temperature. The subject is important, the studies are obviously carefully done, and the interpretation is reasonable. The text is also written in acceptable English, only some typos have to be polished during the edition process (e.g., p.2 l. 83 “a” instead of “an” hydrogen, p. 17, l. 667 “to” instead of “do”).

I only have few comments and suggestions:

-          Please indicate already in the abstract the aim of the paper, namely the study of a soft biomaterial that can be tuned by temperature.

-          Please repeat, at least in the caption of Fig. 7, that the total surfactant concentration is 2 wt%.

-          That the mixture of 2 surfactants leads to a wealth of structures and allows for a deviation from spherical micelles, is not unexpected. Especially, Peter Kralchevsky worked a lot on this subject. Maybe this work should be cited. Compare also: Vera Tchakalova et al., ACS Omega 2023 (https://doi.org/10.1021/acsomega.3c03500).

-          P. 2, l. 48: indeed, 12-HSA is biobased, but not natural, but synthetic. Although used in cosmetics, I am not sure that it can be regarded as non-toxic.

-          In the introduction, much is written about foams and emulsions, whereas the topic of the present manuscript is a homogeneous one-phase system. This is a little bit puzzling.

-          Concerning the mixture of 80% acid and 20% fatty acid salt. Does this mean that 20% can be regarded as surfactant and 80% as an oil so that always swollen micelles would exist and the systems are rather microemulsions than micellar or lamellar aqueous solutions? I think the system is rather special with 20% of salt (compared to the total amount of fatty acids) and 2 % matter in water in total. Is this true?

English ok , only few typos

Author Response

We really thank the reviewer for all his comments. All modifications are indicated in yellow in the main text.

 Suggestions for Authors

  • This is an extremely detailed study of the interesting mixture of two fatty acids that, in aqueous solutions, forms a temperature dependent soft material at room temperature. The subject is important, the studies are obviously carefully done, and the interpretation is reasonable. The text is also written in acceptable English, only some typos have to be polished during the edition process (e.g., p.2 l. 83 “a” instead of “an” hydrogen, p. 17, l. 667 “to” instead of “do”).

We thank the reviewer for these indications and correct it.

I only have few comments and suggestions:

  • -          Please indicate already in the abstract the aim of the paper, namely the study of a soft biomaterial that can be tuned by temperature.

    We correct the abstract in order to clarify the aim of the paper.

  • -          Please repeat, at least in the caption of Fig. 7, that the total surfactant concentration is 2 wt%.

We thank the reviewer for these indications and correct it.

  • -          That the mixture of 2 surfactants leads to a wealth of structures and allows for a deviation from spherical micelles, is not unexpected. Especially, Peter Kralchevsky worked a lot on this subject. Maybe this work should be cited. Compare also: Vera Tchakalova et al., ACS Omega 2023 (https://doi.org/10.1021/acsomega.3c03500).

Thanks to the reviewer, we add some citations of different structures composed of different surfactants which present also different shape than “classical micelles” and which present a transition between different structures (L 577 – 580)

  • -          P. 2, l. 48: indeed, 12-HSA is biobased, but not natural, but synthetic. Although used in cosmetics, I am not sure that it can be regarded as non-toxic.

The reviewer is right, and we removed the sentence on non-toxicity.T

  •   In the introduction, much is written about foams and emulsions, whereas the topic of the present manuscript is a homogeneous one-phase system. This is a little bit puzzling.

Literature on applications of 12-HSA in terms of thermosensitive properties concentrates mainly on foams and emulsions and our introduction reflects this large potential of application. However, we understand the comment of the referee and we add some sentences on application in one phase system. (L101-106)

  •  Concerning the mixture of 80% acid and 20% fatty acid salt. Does this mean that 20% can be regarded as surfactant and 80% as an oil so that always swollen micelles would exist and the systems are rather microemulsions than micellar or lamellar aqueous solutions? I think the system is rather special with 20% of salt (compared to the total amount of fatty acids) and 2 % matter in water in total. Is this true?

The system is constituted of 2wt% of fatty acid and nethanolamine=4* nfatty_acid  (which corresponds to methanolamine=0.8* mfatty_acid  ) thus 1.6wt% in ethanolamine and the rest of water. The counter ion is in large excess and contribute to solvate fatty acids in water for each ratio SA/HSA. In our point of view the system (fatty-acid/ethanolamine) cannot be seen as a microemulsion but more like charged surfactants in water.

Reviewer 2 Report

Manuscript entitled “Aqueous binary mixtures of stearic acid and its hydroxylated counterpart 12-hydroxystearic acid: fine tuning of the lamellar/micelle threshold temperature transition and of the micelle shape” by Almeida and colleges is interesting and great piece of research work. Authors examines the behavior of stearic acid (SA) and 12-hydroxystearic acid (12-HSA) aqueous mixtures as a function of temperature and 12-HSA/SA molar ratio (R). I recommend this work to publish in Molecules after some minor corrections.

My suggestions and have some scientific questions regarding this manuscript are mentioned below:

1. Abstract needs some editing as there are several typological errors for example there is more space between measurements and wide angle…, and in introduction for example so many dots after etc. (line number 38).

Please include the novelty of the work in the introduction and application of this research work. And author already published the similar work “Aqueous Binary Mixtures of Stearic Acid and Its Hydroxylated Counterpart 12-Hydroxystearic Acid: Cascade of Morphological Transitions at Room Temperature” how the present work is different and novel from it? Give a statement in the introduction.

2. This research work of self-assembly and morphological transition can play very important in protein, gene and small molecules (Drug) delivery, in the introduction author should comment on these applications and can cite some work of these delivery systems such as: https://doi.org/10.1016/j.jddst.2023.104403;https://doi.org/10.1016/j.jddst.2022.103699 in addition to that provide more citations on gene and protein delivery as well for example DOI: 10.3390/biomedicines10020493

3. Material and methods: Write the company of origin and manufacturer country for each instrument used in this study for example it is missing for Varian Cary 100 spectrometer and other instruments as well. 

4. Authors should avoid to directly add the DOI in the main text for example “Small angle neutron scattering experiments were performed at the Institut Laue- 643 Langevin – The European Neutron Source (Grenoble, France) on the diffractometer D11 644 (DOI: http://dx.doi.org/10.5291/ILL-DATA.9-11-2041).” There are so many other citations like that. Please review whole manuscript for typological and writing errors.

5. I recommend authors to write few statements why 12-HAS and stearic acid was selected for this study. Provide the rationale behind that. Conclusions need more explanation and next steps of the authors to utilize this study for the drug delivery, foams, stabilizing and other agents.

I recommend this paper to publish in this journal.

Paper is well written and english quality is good.

Author Response

We really thank the reviewer for all his comments. All modifications are indicated in yellow in the main text.

Manuscript entitled “Aqueous binary mixtures of stearic acid and its hydroxylated counterpart 12-hydroxystearic acid: fine tuning of the lamellar/micelle threshold temperature transition and of the micelle shape” by Almeida and colleges is interesting and great piece of research work. Authors examines the behavior of stearic acid (SA) and 12-hydroxystearic acid (12-HSA) aqueous mixtures as a function of temperature and 12-HSA/SA molar ratio (R). I recommend this work to publish in Molecules after some minor corrections.

My suggestions and have some scientific questions regarding this manuscript are mentioned below:

  1. Abstract needs some editing as there are several typological errors for example there is more space between measurements and wide angle…, and in introduction for example so many dots after etc. (line number 38).

We thank the reviewer for this comment and correct such typological errors.

Please include the novelty of the work in the introduction and application of this research work. And author already published the similar work “Aqueous Binary Mixtures of Stearic Acid and Its Hydroxylated Counterpart 12-Hydroxystearic Acid: Cascade of Morphological Transitions at Room Temperature” how the present work is different and novel from it? Give a statement in the introduction.

We thank the reviewer for this comment, and we try to clarify the novelty of this work at the end of the introduction (L 132-135)

  1. This research work of self-assembly and morphological transition can play very important in protein, gene and small molecules (Drug) delivery, in the introduction author should comment on these applications and can cite some work of these delivery systems such as: https://doi.org/10.1016/j.jddst.2023.104403;https://doi.org/10.1016/j.jddst.2022.103699 in addition to that provide more citations on gene and protein delivery as well for example DOI: 10.3390/biomedicines10020493

We add in the main text the application for drug delivery and add citations (L106)

  1. Material and methods: Write the company of origin and manufacturer country for each instrument used in this study for example it is missing for Varian Cary 100 spectrometer and other instruments as well. 

Missing information has been added.

  1. Authors should avoid to directly add the DOI in the main text for example “Small angle neutron scattering experiments were performed at the Institut Laue- 643 Langevin – The European Neutron Source (Grenoble, France) on the diffractometer D11 644 (DOI: http://dx.doi.org/10.5291/ILL-DATA.9-11-2041).” There are so many other citations like that. Please review whole manuscript for typological and writing errors.

We thank the reviewer for this comment and correct it.

  1. I recommend authors to write few statements why 12-HAS and stearic acid was selected for this study. Provide the rationale behind that.

Text has been modified to clarify why HSA and SA have been used (L111-114)

Conclusions need more explanation and next steps of the authors to utilize this study for the drug delivery, foams, stabilizing and other agents.

Text has been modified to explain the potential applications of such HSA/SA mixtures.

I recommend this paper to publish in this journal